# The Trifecta of Single-Cell, Systems-Biology, and Machine-Learning Approaches

**DOI:** 10.3390/genes12071098

**Published:** 2021-07-20

**Authors:** Taylor M. Weiskittel, Cristina Correia, Grace T. Yu, Choong Yong Ung, Scott H. Kaufmann, Daniel D. Billadeau, Hu Li

**Affiliations:** 1Department of Molecular Pharmacology and Experimental Therapeutics, Mayo Clinic College of Medicine and Science, Mayo Clinic, 200 First, Street SW, Rochester, MN 55905, USA; Weiskittel.Taylor@mayo.edu (T.M.W.); correia.cristina@mayo.edu (C.C.); yu.grace@mayo.edu (G.T.Y.); Ung.ChoongYong@mayo.edu (C.Y.U.); kaufmann.scott@mayo.edu (S.H.K.); 2Department of Immunology, Mayo Clinic College of Medicine and Science, Mayo Clinic, 200 First, Street SW, Rochester, MN 55905, USA; Billadeau.Daniel@mayo.edu

**Keywords:** single-cell omics, systems biology, machine learning, single-cell systems biology

## Abstract

Together, single-cell technologies and systems biology have been used to investigate previously unanswerable questions in biomedicine with unparalleled detail. Despite these advances, gaps in analytical capacity remain. Machine learning, which has revolutionized biomedical imaging analysis, drug discovery, and systems biology, is an ideal strategy to fill these gaps in single-cell studies. Machine learning additionally has proven to be remarkably synergistic with single-cell data because it remedies unique challenges while capitalizing on the positive aspects of single-cell data. In this review, we describe how systems-biology algorithms have layered machine learning with biological components to provide systems level analyses of single-cell omics data, thus elucidating complex biological mechanisms. Accordingly, we highlight the trifecta of single-cell, systems-biology, and machine-learning approaches and illustrate how this trifecta can significantly contribute to five key areas of scientific research: cell trajectory and identity, individualized medicine, pharmacology, spatial omics, and multi-omics. Given its success to date, the systems-biology, single-cell omics, and machine-learning trifecta has proven to be a potent combination that will further advance biomedical research.

## 1. Introduction

Single-cell omics describes an ever-increasing arsenal of omic profiling technologies that can interrogate individual cells for their unique genetic and molecular information. The combination of single-cell technologies and systems-biology approaches provides novel opportunities to study biological systems, but the data generated by single-cell technologies also create unique analytical challenges that require powerful computational tools. Single-cell data have low signal-to-noise ratios and high dimensionality compared to traditional bulk omics and are often exceedingly sparse (Figure 1) [1]. The sparsity of scRNA-seq has previously been attributed to technical losses termed “dropout”, but a growing body of evidence suggests that in fact this sparsity is reflective of biological reality [2]. Regardless of the source, computational methods analyzing scRNA-seq must be equipped to handle this sparsity. Another key challenge in single-cell omics is the integration of multiple and multimodal datasets, which requires extensive batch correction [3].

To overcome these challenges, machine-learning techniques have often been applied to single-cell datasets. The efficiency and practicality of machine learning helps to cut through the noise and dimensionality to reveal salient biological insights (Figure 1). Applications of machine learning in biology have been increasing, with particular growth in the use of complex deep-learning models [4]. Deep-learning architectures use multiple layers of networks to reveal high-level features. The two most common subclasses of deep learning are recurrent neural networks (RNN), which progressively feed into themselves recursively, and convolutional neural networks (CNN), which start with convolutional layers that can emphasize input features before feeding into learning layers [5]. These architectures possess unique strengths, making them suitable for differing data types and tasks which have been described in more depth by several review articles [4,5,6,7]. A second unique category of machine-learning approaches includes causal discovery algorithms like probabilistic graphical models which can be used to infer causal relationships, and thus are heavily used in the inference of biological networks [8].

How a machine-learning algorithm accomplishes its tasks is not always known. “Black box” techniques prevent the user from easily understanding learned features, an outcome that can impede biological understanding and proper algorithm training [9]. Importantly, some machine-learning algorithms, like decision trees and regression models, are highly interpretable and thus termed “white box” techniques [9]. Furthermore, concerted efforts have been made to decode “black box” models, particularly in deep-learning applications [9,10,11]. Both “white box” machine-learning algorithms and “black box” interpretation increase interpretability substantially, but many opportunities remain to fully unveil the learned insights of machine-learning models, particularly in the study of biomedicine, where bias and confounding factors need to be addressed [4,7,9]. For many single-cell applications, machine learning is applied in an unsupervised manner because a supervised task is not known or labeled training data are unavailable, and it is mostly used to cluster single cells into meaningful population groups [12]. This overrepresentation of unsupervised machine-learning problems and the general context dependency of biology make validation and significance testing challenging.

The need for advanced computational algorithms in single-cell biology has never been more salient, as the number of techniques has expanded rapidly in the past decade. RNA sequencing is the most common profiling modality at the single-cell level, and technology from platforms such as 10× and Smartseq have continuously pushed the processivity, sensitivity, and affordability of single-cell RNAseq (scRNAseq) [12]. Following the success of scRNAseq, techniques for accessing the genome, epigenome, metabolome, and proteome in single cells have emerged [13]. Thus, it is now possible to access almost all types of omic data at single-cell resolution. Currently, multimodal single-cell omics, where two omic profiles (e.g., proteomics and transcriptomics) are captured for the same cell [14], and spatially resolved techniques are pushing the frontier of possibility [15]. The breadth of single-cell omics available underlines the importance of innovative strategies for advanced data analysis.

Machine learning has been a foundational tool for single-cell analysis from the beginning, but machine learning and single-cell omics are not enough to unveil the full spectrum of mechanistic insights in many applications. Thus, a third pillar is required to push the analytical envelope. Systems biology is a field focused on using computational and mathematical tools to model the systems-wide behavior of biological systems, thus holistically revealing new insights. Given these new technologies and their significant potential for application, a characterization of the utility of the trifecta and remaining gaps is required. Here, we highlight the pros and cons of single-cell omics, machine learning, and systems biology (Figure 1) and describe recent innovations and opportunities in single-cell analysis enabled by this trifecta (Figure 2). We highlight applications of the trifecta in the key fields of cell trajectory and identity, individualized medicine, pharmacology, spatial omics, and multi-omics. Furthermore, we address areas where the trifecta shows great promise and the future paths for its integration in each of these areas.

## 2. Cell Identities and Trajectories

Single-cell technologies have significantly advanced the understanding of evolutionary processes. In particular, single-cell transcriptomic analysis has provided an unbiased approach to query cell trajectories and states. Specifically, single-cell transcriptomics have enabled a unique analysis termed pseudotime, which attempts to place single cells on a plausible developmental trajectory [16]. In a seminal review by Saelens et al., several pseudotime algorithms were compared head to head on complex datasets so that the optimal algorithm for each data type and topology could be determined [17]. Many of these algorithms rely on machine learning to appropriately place cells into a coherent trajectory [17]. Although specific methodologies for trajectory inference are outside the scope of this review, key insights from these trajectories can be gleaned by layering systems information on top of pseudotime trajectories. These annotations include sample identity, cluster membership, and gene expression [18]. This layering approach has provided ample new insights, but future approaches need to integrate systems biology earlier to produce more mechanistically faithful trajectories while simultaneously revealing biological insights. Similarity matrix-based optimization for single-cell analysis (SoptSC) accomplishes this by simultaneously inferring pseudotemporal clusters and cell–cell communication networks using unsupervised machine-learning methods [10]. By accomplishing these tasks in an integrated pipeline, the inferred trajectories were more coherent with underlying biology such as the asynchronous development of the myeloid compartment in murine hematopoiesis [19]. Such integration of systems biology and machine learning into coherent pipelines for developmental inference will certainly lead to more insights in the future (Figure 2).

Trajectory analysis of single-cell data has often revealed previously unknown intermediate cell states. Prior to this analysis, cell states were determined by looking for the presence of known markers, thus representing a binary classification. Now we can appreciate the intermediate states that are poorly delineated with known cell markers. With these states identified, systems biologists can now model such states to better understand the biology occurring at crucial transitions [20,21]. For example, the probabilities that granulocyte–monocyte progenitors would differentiate were modeled using single-cell experimentation and Bayesian computing, and this modeling elucidated the time dependency and probabilities of this transition [21]. Cellular identity and transitions are currently a major focus of evolutionary single-cell studies, but more work is needed. Wagner et al. proposed in their review that single cells possess multiple “vectors of cellular identity” which encapsulate the functions and cellular circuitry that in aggregation lead to identity [22]. Advanced machine learning will likely be needed to deconvolute these identity vectors due to the high dimensionality of single-cell data and the numerous identity and transitional vectors that remain to be uncovered (Figure 2). Reconstructing cell identity vectors is not currently possible using existing pipelines, but with careful construction using machine learning such pipelines should be feasible.

## 3. Pharmacology

Pharmacology has been revolutionized by omic data characterization. Systems biologists have been an integral part of this revolution, which has led to the new discipline of systems pharmacology that has since produced high-value resources and discoveries [23,24,25]. The connections between drugs, diseases, and biological signatures have been mapped using hierarchical clustering on the bulk gene expression profiles of drugged perturbed cell lines, thus establishing a “connectivity map” (CMAP) of pharmacological and disease mechanisms [26]. Several other resources like CMAP have emerged from the efforts of systems pharmacologists to reveal new mechanisms and coordinate large amounts of data using advanced mathematics and machine learning (Figure 2) [27,28,29]. Furthermore, many research groups have ventured into predictive pharmacology, where they identify putative drug targets, disease response to a therapy in question, or side effects. Systems networks constructed from transcriptomic data like CMAP have been utilized to predict both new drug targets and existing drug side effects. Taking this a step further, the multi-omic late integration framework uses deep neural networks to predict chemotherapy agent response from multi-omics and has demonstrated that transfer learning is a successful strategy for increasing prediction accuracy in pharmacology [30]. The further development of pharmacotherapy predictive systems working from single-cell data presents several key opportunities. First, drugs often target and create side effects in specific cell types, which makes deconvoluting these populations molecularly imperative (Figure 2). Second, in many diseases, a drug must combat disease-causing cells that are part of a heterogenous population, particularly in cancer and infectious disease (Figure 2). Thus, single-cell techniques will likely increase the accuracy of predictive systems algorithms because they allow for increased cell type specificity and characterization of heterogeneity.

Most systems biology research has focused on scRNA-seq, as it is the most ubiquitous single-cell technique. However, several niche single-cell technologies have extraordinary potential in pharmacology and can be combined with systems-biology and machine-learning approaches for maximum benefit. Single-cell biofluorescence analysis provides detailed and high-throughput screening and, when analyzed using deep neural networks, can reveal the mechanisms of action of screened drugs [31]. In another application of biofluorescent drug screening and machine learning, the idTRAX algorithm was able to find cancer-selective kinase inhibitors [32]. In a final example, machine learning was used to classify phenotypic variations caused by drugs from three-dimensional screening data of leukemia cells [33]. These applications demonstrate how single-cell screening paired with machine learning can provide biological insights beyond the drug efficacy readouts seen in past screening strategies.

## 4. Spatial Omics

As single-cell frontiers are increasingly being explored, new technology has emerged that allows for the retention of spatial information when probing omics data. In standard single-cell omics protocols, cells from the sampled tissue are separated to allow for barcoding and preprocessing. Recently, advances in in-situ hybridization (ISH) techniques, spatial dissection, and spatial barcoding have allowed for the simultaneous capture of RNA abundance data while retaining spatial architecture [15,34]. These studies have a high degree of relevance for understanding a diverse range of topics, including embryonic development, normal tissue organization, and tumor niche architecture. To date, these pipelines experience limitations in the number of genes they are able to probe, scalability, and resolution, but consistent progress is being made in overcoming these hurdles (Figure 2) [34].

A host of computational algorithms has grown around spatial transcriptomics for data integration and analysis. Seurat and other pipelines use machine learning to match ISH data to scRNAseq from the dissociated tissue, thus allowing for spatial assignment of single-cell transcriptomes [35]. CSOmap takes an alternative approach and constructs a spatial map de novo using a ligand–receptor network and dimensional reduction [36]. The spatial coordinates and scRNAseq can be further understood using downstream analysis pipelines that reveal receptor–ligand pairs, neighborhood properties, and spatial expression patterns [37,38]. Each of these programs reduces the barrier to entry for spatial transcriptomics, which will allow this to become a more routine analysis.

Systems-biology analyses will need to grow with the field of spatial omics. Some concepts from older systems approaches can be adapted to handle spatial data, just as bulk concepts were adapted to single cells. However, like the bulk of the single-cell transition, many will not be transferable because they lack the high throughput needed to handle the increased dimensionality of spatially resolved omics (Figure 1). Nevertheless, systems-biology analysis is needed to understand the emergent properties in spatial data, and in particular to better elucidate spatial signaling and the spatial dynamics of regulatory networks (Figure 2). Machine-learning and dimensionality-reduction techniques will likely prove to be the workhorse tools of this effort, as they possess more throughput than statistically based processes. Thus, machine learning and systems biology will likely reveal exciting insights in the spatial omics realm, as we have seen with single-cell omics as a whole.

## 5. Multi-Omic Characterizations

As described in the introduction, single-cell technology can now concurrently profile pairs of omic modalities [39]. Integrative analysis pipelines such as MOFA and LIGER allow for cells to be clustered based on features from both modalities using machine learning and dimensionality reduction [40,41]. Integrative analysis that considers both omic profiles at once is key, because cluster identity is then based off both levels of omic analysis. Advanced mechanistic analysis of multi-omics at the systems level remains difficult even for bulk datasets (Figure 2) [42]. The key challenge of bulk multi-omics remains the variance in scale, noise, and quantitative ability [42]. Single-cell analysis adds additional obstacles of higher noise and dimensionality.

Thus far, systems biology and machine learning in multi-omics have primarily been applied to predictions of cancer clinical variables (Figure 2). Ma et al. integrated multi-omic bulk data with molecular interaction networks to predict clinical variables like survival; by adding domain knowledge, e.g., molecular integration networks such as STRING and Reactome data, as inductive biases, they prevented overfitting of their deep-learning algorithms [43]. Ramazzoti et al. focused on creating multi-omic-based cancer subtypes using multikernel learning. These new subtypes correlated with clinical outcomes and recapitulated known and novel omic markers [44]. These hallmark studies demonstrated the utility of multi-omics in the bulk setting; further insights will undoubtedly be gained from similar studies in single-cell contexts. In both the bulk and single-cell settings, however, innovation needs to be applied further to reveal more systems-informed mechanisms by operating at different omic levels and with several omic modalities.

## 6. Individualized Medicine

Individualized medicine seeks to tailor treatment to each patient. The advent of omic analysis has propelled this field into an entirely new era. Sequencing and omic techniques give unparalleled insights into each patient’s cellular environments, and single-cell techniques allow us to further characterize the heterogeneity and microenvironments seen in each patient. In bulk omics, machine-learning and systems-biology approaches have primarily focused on identifying disease variants in genomic sequencing data [45]. Fewer studies have tackled precision medicine mechanistically. Deep learning has been shown to accurately classify disease-causing splice site mutations that, when annotated with protein binding and disease data, reveal disease mechanisms in individual patients [46]. Zhou et al. took this a step further by predicting both disease risk and expression changes caused by mutational variants [47]. More clinically based studies have used machine learning on bulk omics to classify cancer subtypes and stratify patients, but little was revealed by these studies about causal mechanisms or new therapeutic opportunities (Figure 2) [48,49].

To date, systems-biology-enabled mechanistic investigation of individual patients has not transferred into the single-cell realm. The scarcity and high dimensionality of single-cell data renders many of the prior precision medicine algorithms impractical for single-cell applications (Figure 1). However, single-cell approaches represent a significant opportunity in the precision medicine space because they can provide thousands of data observations (cells) per patient, which is often required for machine-learning algorithms. In this way, interpretable machine-learning algorithms could be constructed based purely on a single patient’s data, and the learned features of these algorithms could reveal individualized disease mechanisms (Figure 2).

## 7. The Future of the Trifecta

The era of single-cell systems biology is still in its infancy, but the promise remains immense. The dimensionality, sparsity, technical and biological noise, and diversity of single-cell omic profiles requires novel advanced computing strategies. Thus far machine learning techniques have proven to be a major avenue for overcoming these hurdles (Figure 1). To truly go beyond outcomes and statistical correlates, a systems perspective that emphasizes mechanistic insights is required. This perspective has been incredibly successful with bulk omic assays; the next frontier is to create a similar variety of approaches for single-cell data.

To illustrate the impact of this trifecta of machine-learning, single-cell omics, and systems biology, we have discussed five key research areas herein. The trifecta has been applied to differing extents in each of these areas (Figure 2). Some, like pharmacology and individualized medicine, still primarily pull from bulk datasets, but use machine learning and systems biology extensively. These fields thus lack a high-resolution perspective that displays the diversity of cellular phenotypes (Figure 1). In contrast, spatial omics and multi-omics researchers frequently use machine learning to process single-cell data, but mechanistic and biological meaning is not often explored at the systems level (Figure 1). Adding systems biology with enhanced interpretability of deep- or machine-learning algorithms will push the mechanistic learning of the fields forward substantially. Cell trajectory and identity studies have made strong use of all three methods; accordingly, this is one of the most advanced areas of single-cell biology. For each of these sections, we have highlighted a variety of works that display the state of the art and propose a path forward for new innovations that would require the discussed trifecta (Figure 2).

## 8. Conclusions

As single-cell techniques expand in breadth and availability, the sea of big data grows deeper and more mysterious. Machine learning can recover pearls of predictive variables and correlative associations, but the mechanism behind these instances is left unknown. However, with recent endeavors to “open the black box” of machine learning, we believe such obstacles will be resolved in near future. With machine-learning-enabled systems biology, we have armed analytical approaches to find both the “what” and the “why” behind biological phenomena within the depths of single-cell omics. Computational researchers must continue to utilize this trifecta to reveal meaningful emergent properties of cellular systems at an ever-increasing resolution.

## Figures and Tables

**Figure 1 genes-12-01098-f001:**
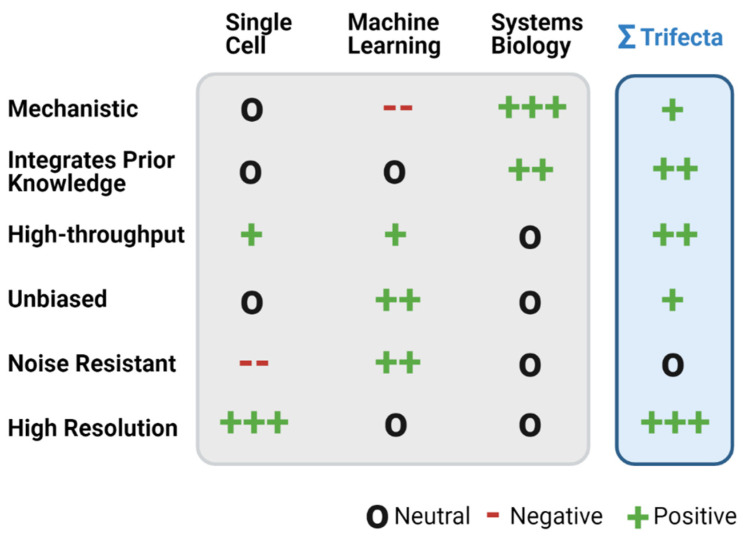
Strengths and weaknesses of the trifecta and their combinatorial benefits. A summary chart of the strengths (green) and weaknesses (red) of machine-learning, systems-biology, and single-cell omics in key areas of challenge or need (rows). The blue trifecta column highlights how combined negatives are countered and more needs are met. (Created with BioRender).

**Figure 2 genes-12-01098-f002:**
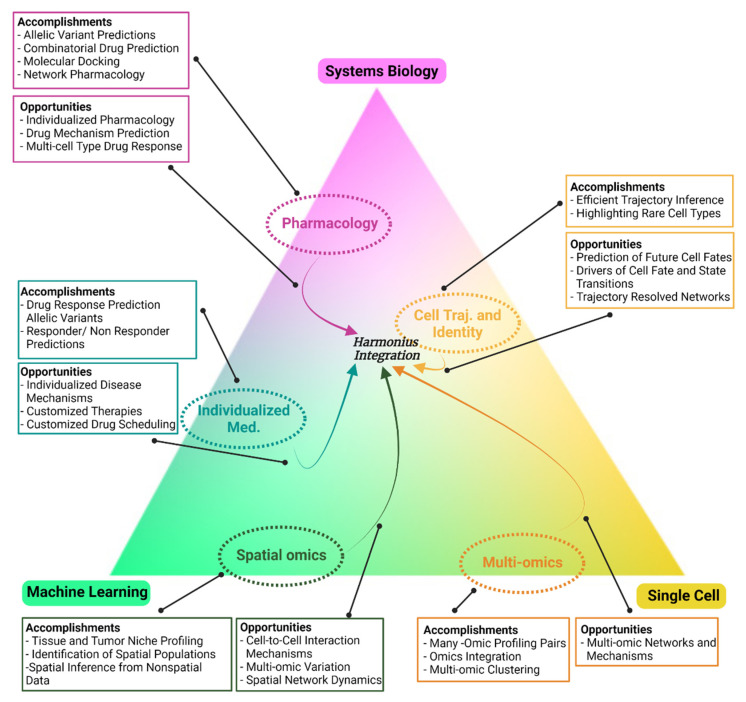
The success and opportunities of key fields within the trifecta. The current degree of integration of each key research areas with single-cell data, systems biology, and machine learning. Integration is represented by ovals’ proximity to each corner of the trifecta. The center represents equal and wholistic integration, which we suggest will be of great utility. Accomplishments and opportunities in each key field are listed in the matching colored boxes. (Created with BioRender.).

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
