# Peer review of "The Trifecta of Single-Cell, Systems-Biology, and Machine-Learning Approaches"

_genes, 2021, doi:10.3390/genes12071098_

Round 1
Reviewer 1 Report
The authors have attempted to combine approaches of growing interest in the study of mechanisms operating in living beings, such as those provided by single-cell sequencing technology, Systems Biology and Machine/Deep Learning algorithms. However, the manuscript presents important weaknesses that must be overcome:
- There is no clear and concise description of what single-cell sequencing technology consists of, what is the way to proceed in Systems Biology and what are the most commonly used Machine/Deep Learning models in Biology, why and what for.
- Regarding Deep Learning, there is not even mention of the two most important classes of neural networks and the implications of using one or the other: Convolutional Neural Network (CNN) and Recurrent Neural Network (RNN).
- No mention is made of Causal Discovery algorithms, such as Probabilistic Graphical Models, despite their importance for learning networks, such as Gene Regulation Networks, from different datasets:
Nagarajan, R., Scutari, M., & Lèbre, S. Bayesian Networks in R with Applications in Systems Biology 2013. New York, NY Springer-Verlag.
- The authors do not discuss whether single-cell sequencing datasets are zero-inflated or not, what possible solutions exist (if needed) and what consequences these solutions could have on the biological conclusions drawn:
Svensson, V. (2020). Droplet scRNA-seq is not zero-inflated. Nature Biotechnology, 38(2), 147-150.
- Machine Learning algorithms are described as "black boxes". However, many Machine Learning algorithms are interpretable "white boxes", such as Decision Trees. Deep Neural Networks are the ones that could be considered "black boxes", but there is a growing trend to develop Explainable or Interpretable Neural Networks:
Angelov, P., & Soares, E. (2020). Towards explainable deep neural networks (xDNN). Neural Networks, 130, 185-194.
Fortelny, N., & Bock, C. (2020). Knowledge-primed neural networks enable biologically interpretable deep learning on single-cell sequencing data. Genome biology, 21(1), 1-36.
- Lines 59-62: there is a lack of citations that adequately support the argumentation presented.
- On what basis do the authors define the values used in Figures 1 and 2? These values are not adequately justified.
Author Response
The authors have attempted to combine approaches of growing interest in the study of mechanisms operating in living beings, such as those provided by single-cell sequencing technology, Systems Biology and Machine/Deep Learning algorithms. However, the manuscript presents important weaknesses that must be overcome:
We thank the Reviewer and greatly appreciate the helpful detailed comments on our manuscript titled “The Trifecta of Single Cell, Systems Biology, and Machine Learning Approaches” and implemented their suggestions which we found to enhance our manuscript. A point-by-point response is detailed below:
- There is no clear and concise description of what single-cell sequencing technology consists of, what is the way to proceed in Systems Biology and what are the most commonly used Machine/Deep Learning models in Biology, why and what for.
- Defining Single Cell Technology: We have now clarified how we define single cell technologies (see lines 27-29) We have attempted to demonstrate a path forward from a systems and machine learning perspective by describing a few representative single cell technologies (lines 69-80), but appreciate that the added definition helps clarify the scope of single cell technologies we will be discussing.
- Common Machine Learning Methods: In systems biology, machine learning methods such as linear regression, decision trees, and support vector machines are used in unconventional ways to derive biological and mechanism insights in disease. In this paper we focused on single cell omics data and methodologies that cross the boundaries of systems biology and machine learning to derive novel biological knowledge. In this way and because the tasks and use cases are diverse, a most common machine learning method is hard to pinpoint. More conventional classification applications are out of the scope of this review however we have elaborated on these approaches within the text and highlighted pertinent citations where readers can learn more about these specifically (lines 49-51).
- Regarding Deep Learning, there is not even mention of the two most important classes of neural networks and the implications of using one or the other: Convolutional Neural Network (CNN) and Recurrent Neural Network (RNN).
- No mention is made of Causal Discovery algorithms, such as Probabilistic Graphical Models, despite their importance for learning networks, such as Gene Regulation Networks, from different datasets:
Nagarajan, R., Scutari, M., & Lèbre, S. Bayesian Networks in R with Applications in Systems Biology 2013. New York, NY Springer-Verlag.
- The authors do not discuss whether single-cell sequencing datasets are zero-inflated or not, what possible solutions exist (if needed) and what consequences these solutions could have on the biological conclusions drawn:
Svensson, V. (2020). Droplet scRNA-seq is not zero-inflated. Nature Biotechnology, 38(2), 147-150.
- Addition of pertinent topics: We thank the Reviewer for the helpful references on causal discovery algorithms (lines 51-53), CNN/ RNN (lines 44-51), and zero inflation (lines 33-36) which have been added to the introduction. We discussed zero inflation in non-technical terms to better suit the intended audience so the phrase “zero-inflation” is not explicitly used but have described the core issue.
- Machine Learning algorithms are described as "black boxes". However, many Machine Learning algorithms are interpretable "white boxes", such as Decision Trees. Deep Neural Networks are the ones that could be considered "black boxes", but there is a growing trend to develop Explainable or Interpretable Neural Networks:
Angelov, P., & Soares, E. (2020). Towards explainable deep neural networks (xDNN). Neural Networks, 130, 185-194.
Fortelny, N., & Bock, C. (2020). Knowledge-primed neural networks enable biologically interpretable deep learning on single-cell sequencing data. Genome biology, 21(1), 1-36.
- White verses black box and interpretability: We appreciated the reviewer for drawing attention to the growing interpretability of machine learning. We have revised our discussion to be more inclusive of “white box” and interpretation algorithms (lines 54-63).
- Lines 59-62: there is a lack of citations that adequately support the argumentation presented.
- Citations in Introduction: We have added clarification that this paragraph prefaces the discussion that follows in the paper, and that the citations in the subsequent text support our belief that the trifecta of approaches is of high value in biology and medicine (lines 81-94).
- On what basis do the authors define the values used in Figures 1 and 2? These values are not adequately justified.
- Figures: The intent and message of the figures has been revised and defined earlier in the piece as to state their value more clearly to the reader (lines 88-90). The figure titles have also been edited to be more explicit as suggested (lines 287-290 and 292-296).
Reviewer 2 Report
Weiskittel et al. provide a review on the latest developments on the integration of single-cell analysis, systems biology and mashine learning approaches. Regarding this integrative approach, they report on recent advances in different research fields, including cell identity/trajectory, pharmacology, spatial -omics, multi -omics and individualized medicine. The authors suggest future developments/improvements, and discuss challenges that need to be overcome to push the field further.
The review is well written, concise and comprehensive. I only have very few and small comments.
Can you elaborate more on the drawbacks of mashine learning approaches (i.e., black box approaches)?
The reader may not be familiar with the systems biology approach of research. Could you spend a few words about the general strategy?
Figure 2: I could not read the word “Spatial-omics” in my print-out version. You may change the color for better differentiation (currently, it is green on green).
Line 86: What does “SopSC” mean? Maybe you need to introduce the abbreviation.
Line 226-229. Please remove the placeholder text.
A couple of unnecessary spaces and signs: line 39, 115, 124, 144, 162, 196, 205, 210, 214, 247, 271
Round 2
Reviewer 1 Report
I thank the authors for the great effort in attempting to address the state-of-the-art of constantly improving technologies such as single-cell sequencing technology and Machine/Deep Learning algorithms, as well as for adequately addressing all the changes and corrections I suggested in the previous version of the manuscript. The manuscript has been considerably improved and covers the fundamental aspects of how these technologies can be coupled with Systems Biology to produce new biological knowledge, so I recommend publishing the manuscript in its current form.